# MicroRNA expression profiles in molecular subtypes of clear-cell renal cell carcinoma are associated with clinical outcome and repression of specific mRNA targets

Annelies Verbiest[1,2], Vincent Van Hoef[3], Cristina Rodriguez-Antona[4,5], Jesús García-Donas[6,7], Osvaldo Graña-Castro[8], Maarten Albersen[9], Marcella Baldewijns[10], Annouschka Laenen[11], Eduard Roussel[9], Patrick Schöffski[1,2], Agnieszka Wozniak[1,2], Stefano Caruso[12], Gabrielle Couchy[12], Jessica Zucman-Rossi[12], Benoit Beuselinck[1,2]*

1 Department of General Medical Oncology, Leuven Cancer Institute, University Hospitals Leuven, Leuven, Belgium, 2 Department of Oncology, Laboratory of Experimental Oncology, KU Leuven, Leuven, Belgium, 3 Bioinformatics Expertise Center, VIB, Leuven, Belgium, 4 Hereditary Endocrine Cancer Group, Human Cancer Genetics Programme, Spanish National Cancer Research Centre, Madrid, Spain, 5 Centro de Investigación Biomédica en Red de Enfermedades Raras, Madrid, Spain, 6 Oncology Unit, HM Hospitales—Centro Integral Oncologico HM Clara Campal, Madrid, Spain, 7 Spanish Oncology Genitourinary Group, Madrid, Spain, 8 Bioinformatics Unit, Structural Biology and Biocomputing Programme, Spanish National Cancer Research Centre, Madrid, Spain, 9 Department of Urology, University Hospitals Leuven, Leuven, Belgium, 10 Department of Imaging and Pathology, KU Leuven, Leuven, Belgium, 11 Biostatistics and Statistical Bioinformatics Center, KU Leuven, Leuven, Belgium, 12 Inserm, UMR-1162, Génomique fonctionnelle des tumeurs solides, IUH, Paris, France

* benoit.beuselinck@uzleuven.be

**Data Availability Statement:** miRNA-seq data have been deposited in the ArrayExpress database

## Abstract

Clear-cell renal cell carcinomas (ccRCC) can be divided into four transcriptomic subtypes, two of which have a favorable and two an unfavorable prognosis. To assess mechanisms driving these subtypes, we investigated their miRNA expression patterns. miRNAs are master regulators of mRNAs, that are widely deregulated in cancer. Unsupervised clustering in our dataset (n = 128) and The Cancer Genome Atlas (TCGA) validation set identified two distinct miRNA clusters that overlapped with the transcriptomic subtypes, underscoring the validity of these subtypes on a multi-omics level and suggesting a driving role for miRNAs. Discriminatory miRNAs for the favorable subtypes repressed epithelial-to-mesenchymal transition, based on gene set enrichment analysis and target-mRNA expression levels. Strikingly, throughout the entire dataset, miRNAs associated with favorable subtypes were also associated with longer overall survival after diagnosis, and miRNAs associated with unfavorable subtypes with shorter overall survival (Pearson r = -0.54, p<0.0001). These findings indicate a general shift in miRNA expression between more and less aggressive tumors. This adds to current literature, which usually suggests only a small subset of miRNAs as markers of aggressive disease. In conclusion, this study reveals distinct mRNA expression patterns underlying transcriptomic ccRCC-subtypes, whereby miRNAs associated with favorable subtypes counteract epithelial-to-mesenchymal transition. There is a general shift in miRNA expression in ccRCC, between more and less aggressive tumors.

at EMBL-EBI (www.ebi.ac.uk/arrayexpress) under accession number E-MTAB-9380. All clinical outcome data are available in S1 Table.

**Funding:** The author(s) received no specific funding for this work.

**Competing interests:** I have read the journal's policy and the authors of this manuscript have the following competing interests: Benoit Beuselinck received consultancy fees from Amgen, Ipsen, Pfizer and Novartis and institutional research grants from Bristol-Myers Squibb and Ipsen. Patrick Schöffski has received consultancy fees as well as institutional research grants from Merck and Exelixis. The other authors have no conflicts of interest to declare. This does not alter our adherence to PLOS ONE policies on sharing data and materials.

## Introduction

miRNAs are small non-coding RNAs that repress gene expression. They bind to their complementary sequence on mRNAs, thus inducing the mRNA's degradation or preventing translation. miRNAs regulate key cellular processes in an intricate way, with each miRNA having up to hundreds of mRNA targets, and each mRNA being targeted by several miRNAs [1]. In general, regulation by miRNAs aids to maintain cells in a differentiated state. Conversely, cancer cells are hallmarked by a global downregulation of miRNA expression, which allows them to develop a more undifferentiated phenotype [2,3]. Apart from this widespread loss of miRNA expression in cancer, individual miRNAs may be upregulated and can play either an oncogenic or a tumor suppressor role. To add another layer of complexity, this role can vary depending on the tumor type and specific circumstances. In clear-cell renal cell carcinoma (ccRCC) as well as in other cancer types, several miRNAs have been associated with prognosis, but often with inconsequent or contradictory results in different series [4–9].

On the mRNA level, we have previously described four molecular subtypes in advanced ccRCC, which are based on unsupervised clustering of whole transcriptome data [10]. Two of these subtypes have a favorable prognosis: ccrcc2 (45%, highly angiogenic) and ccrcc3 (5%, resembling normal kidney). The two others are associated with an aggressive stem cell-like phenotype and poor outcomes, but have different immune phenotypes: ccrcc1 (30%, immune-cold) and ccrcc4 (20%, inflamed). Similar mRNA-based molecular subtypes have been described by other groups [4,11,12]. Although these subtypes also display different mutation profiles, they are not fully determined by specific genomic alterations. Therefore, they are also driven by epigenetic regulatory processes that remain incompletely understood. Methylation status plays a role, with the poor prognostic ccrcc1 and ccrcc4 tumors being globally hypo-methylated. On the other hand, as miRNAs are master regulators of gene expression, they are also excellent candidates to partially explain the different gene expression profiles between the subtypes [10]. Indeed, previous clustering of miRNA data in the The Cancer Genome Atlas (TCGA) cohort has revealed four miRNA clusters, two of which overlapped with transcriptomic mRNA clusters that were also described by TCGA [4]. Among key discriminatory miR-NAs, miR-21 was strongly correlated with poor prognosis and repressed targets in the Von Hippel Lindau—Hypoxia Inducible Factor 1-alpha (VHL–HIF1A) axis, which is critically deregulated in ccRCC. In other tumor types as well, miRNA expression has been shown to differ between molecular subtypes. For instance, in colorectal cancer, the poor prognostic 'mesenchymal' molecular subtype is associated with downregulation of miRs 194, 200b, 203 and 429 [13]. In bladder cancer, the less aggressive 'luminal' molecular subtype is associated with downregulation of miR-99a, miR-100, miR-145 and miR-125b [14].

In this study, we performed a miRNome analysis of primary tumors from a large series of metastatic ccRCC patients. We aimed to identify miRNAs driving mRNA molecular subtypes, to assess their repression of mRNA-targets and to determine the association between subtype-specific miRNA expression and clinical outcome.

## Results

We included 128 patients for miRNome and clinical analysis, and determined the molecular subtype in 95 cases where fresh frozen tissue was available. Forty-five tumors had a favorable ccrcc2_3 subtype (39 ccrcc2, 6 ccrcc3), 50 had an unfavorable ccrcc1_4 subtype (24 ccrcc1, 26 ccrcc4). Of the 16 paired normal and tumor samples, the molecular subtype determined for 11 cases. Patient characteristics are summarized in Table 1. miRNA-seq data have been deposited in the ArrayExpress database at EMBL-EBI (www.ebi.ac.uk/arrayexpress) under accession number E-MTAB-9380.

**Table 1. Patient characteristics.**

| All patients | 128 | |
|---|---|---|
| Male | 86 | 67% |
| Median age at diagnosis | 62 yrs | |
| Synchronous metastases | 62 | 48% |
| Median DFI if metachronous | 27 mo | |
| Median OS after diagnosis | 49 mo | |
| Median OS after stage IV | 34 mo | |
| First line targeted therapy[A] | | |
| Sunitinib | 68 | 53% |
| Pazopanib | 31 | 24% |
| Sorafenib | 11 | 9% |
| Temsirolimus | 9 | 7% |
| Nivolumab-Ipilimumab | 6 | 5% |
| Other | 3 | 2% |
| IMDC risk group | | |
| Favorable | 14 | 11% |
| Intermediate | 81 | 63% |
| Poor | 33 | 26% |
| ccrcc1 to -4 molecular subtype | | |
| ccrcc1 | 24 | 25% |
| ccrcc2 | 39 | 41% |
| ccrcc3 | 6 | 6% |
| ccrcc4 | 26 | 27% |
| Unknown | 33 | |

[A] All patients were treated for metastatic disease. No patients received adjuvant therapy.

DFI = disease free interval (months); OS = overall survival (months); IMDC = International Metastatic ccRCC Database Consortium

In the TCGA validation set, combined miRNA and mRNA expression data with ccrcc1 to -4 molecular subtype were available for 107 samples.

## Exploratory unsupervised clustering reveals two miRNA clusters that overlap with molecular subtypes

We evaluated miRNA expression patterns across all samples in an unbiased way, using principal component analysis and unsupervised clustering. These analyses revealed a marked differentiation of tumoral *vs* normal kidney samples (Fig 1A–1C). As shown in the heatmap, the tumor cluster further separated into two clusters that mainly overlapped with the molecular subtypes. Unsupervised clustering using only the tumor samples, separated the samples into one cluster that consisted for 81% of the unfavorable ccrcc1_4 tumors, and a second cluster that consisted for 62% of the favorable ccrcc2_3 tumors (p = 0.0005). These findings were validated on the TCGA dataset, where we identified two miRNA clusters with 66% and 91% overlap with favorable and unfavorable molecular subtypes respectively (p = 4.7e-9) (S1 Fig). The heterogeneity of miRNA expression was higher in the unfavorable ccrcc1_4 subtypes. We also compared 11 tumor samples with their matched normal kidney (7 ccrcc2_3, 4 ccrcc1_4). The miRNAs that were differentially expressed with normal kidney, largely overlapped between ccrcc2_3 and ccrcc1_4 tumors (Fig 1D). However, the extent to which they were differentially expressed between normal kidney and tumors, differed between ccrcc2_3 and ccrcc1_4.

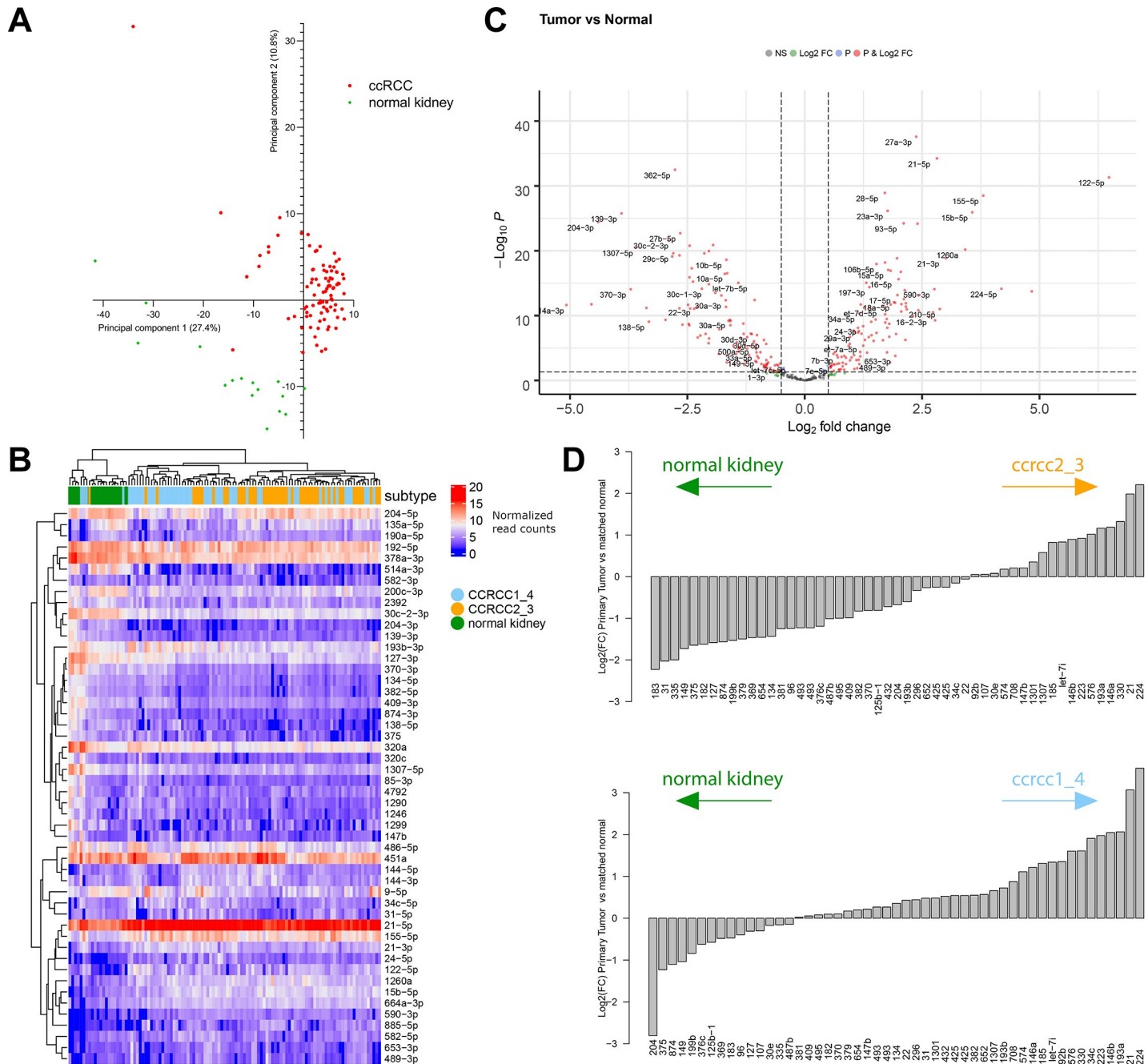

**Fig 1. Exploratory analyses of all samples (tumor and normal kidney combined, Leuven dataset).** These showed a marked separation of normal *vs* tumoral kidney samples. **(A)** Principal component analysis (n = 111). **(B)** Unsupervised cluster analysis of the 50 most variable miRNA (n = 111). **(C)** Volcano plot (n = 95). **(D)** Significantly differently expressed miRNAs between matched normal and tumoral samples (n = 16). In panels C and D, higher positive fold changes indicate higher expression in tumor samples and lower negative fold changes indicate higher expression in normal kidney.

## Molecular subtype-specific miRNA expression is involved in pathways that regulate tumor biology

We identified the miRNAs that displayed significantly different expression levels between the ccrcc2_3 and ccrcc1_4 subtypes (Fig 2A, Table 2, S1 Table). The favorable and unfavorable subtypes had markedly different expression of several miRNAs, many of which have been

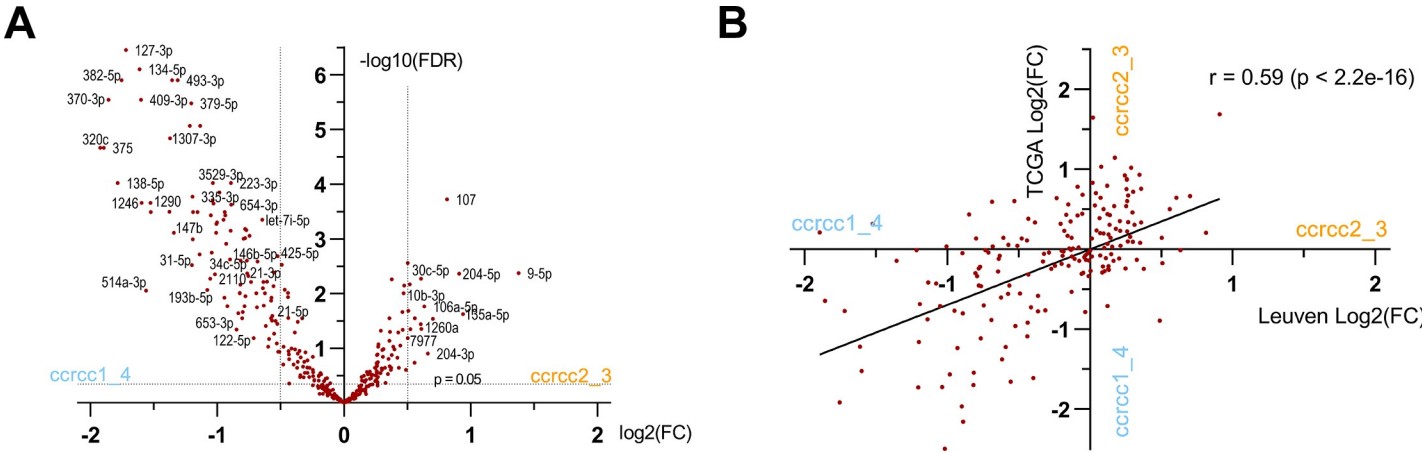

**Fig 2. Illustration of subtype-specific miRNA expression.** Ccrcc2_3 and ccrcc1_4 tumors showed distinct miRNA expression patterns. **(A)** Volcano plot of miRNAs with subtype-specific expression (Leuven dataset, n = 95). **(B)** Subtype-specific miRNA expression in the Leuven dataset was validated on TCGA by Pearson correlation (r = 0.59) (n = 95 Leuven and n = 107 TCGA). The correlation was performed on 189 miRNAs that were shared between the Leuven and TCGA dataset. FC = fold change; FDR = false discovery rate.

widely associated with relevant pathways in cancer biology (Table 3). These findings were validated on the TCGA data: the subtype-specific expression of miRNAs, as quantified by fold change, was strongly correlated between our dataset and the TCGA with a Pearson coefficient of r = 0.59 (p < 2.2e-16) (Fig 2B).

In order to identify functional miRNA-mRNA interactions of subtype-specific miRNAs, we correlated these miRNAs with their predicted mRNA targets using the TCGA dataset (full transcriptome data were available for TCGA, but not for our own dataset). Among several repressed mRNAs, 29 targets of 12 miRNAs displayed significantly different expression levels between the molecular subtypes, with log2(FC) ≥ 0.5 or ≤ -0.5 in ccrcc2_3 *vs* ccrcc1_4 (Table 2, Fig 3B). These specific mRNAs play an important role in the epithelial-to-mesenchymal transition observed in tumor cells: many of the targets that were repressed in the favorable ccrcc2_3 subtypes are mRNAs that promote tumor invasiveness and metastasis (*MMP3*, *MMP9*, *AURKB*, *FBN2*, *ITGB4*, *SPDEF*, *F2*, *RUNX2*, *CDCP1*, repressed by miRs 204-5p, 30c-5p, 135a-5p and 106a-5p). Contrarily, miRNA 96-5p, which was upregulated in the unfavorable ccrcc1_4 subtypes, repressed Myosin VIIa- and Rab-Interacting Protein (*MYRIP*), the latter being involved in anchoring and scaffolding processes to retain cell structure [15].

We then performed KEGG gene set enrichment analysis to identify cellular pathways affected by subtype-specific miRNA expression. Although these results do not fully capture the complex multi-level regulation by miRNAs and should be interpreted with caution, they did show a relative upregulation of several pathways that promote cell structure and adhesion in the favorable ccrcc2_3 subtypes (S2 Fig). The names of these KEGG pathways are: "cell adhesion molecules", "gap junction", "regulation of actin cytoskeleton", "tight junction", "focal adhesion" and "adherens junction".

## MiRNA expression is correlated with both overall survival and molecular subtype

To assess the translational relevance of our findings we tested the correlation of miRNA expression with overall survival (OS) using Cox regression. Our hypothesis was that the limited subset of miRNAs with marked subtype-specific expression would also show a prognostic value. However, strikingly, OS analysis revealed a much more remarkable association:

**Table 2. List of miRNAs that exhibited subtype-specific expression (FDR < 0.05) and that were either significantly associated with OS (Leuven dataset) or suppressed an mRNA target that was downregulated with a fold change of < 0.5 compared to the other molecular subtype (TCGA dataset).** miRNAs with higher expression in ccrcc1_4 subtypes were systematically associated with shorter OS, and those with higher expression in ccrcc2_3 with longer OS. Downregulated mRNA targets are listed below the miRNA. Only experimentally validated mRNA targets were considered (sources: TarBase, TargetScan Human, miRecords, Inguinity Expert Findings). mRNA subtype-specific fold changes are illustrated in Fig 3B.

**miRNA upregulated in ccrcc2_3 tumors**

| | log2(FC) | FDR | HR(OS) | FDR |
|---|---|---|---|---|
| **101-3p** | 0.43 | 0.051 | 0.91 | 0.54 |
| *PTGS2* | | | | |
| **23c** | 0.46 | 0.046 | 0.89 | 0.23 |
| *PLAU* | | | | |
| **7977** | 0.52 | 0.045 | 0.69 | 0.00009 |
| **1260a** | 0.61 | 0.044 | 0.70 | 0.0002 |
| **30c-5p** | 0.61 | 0.0053 | 0.80 | 0.074 |
| *F2, MLLT11, CDCP1, SYT4, RUNX2, SLC7A1* | | | | |
| **106a-5p** | 0.63 | 0.017 | 0.91 | 0.46 |
| *MMP3. CXCL8. RUNX1* | | | | |
| **204-5p** | 0.91 | 0.004 | 0.73 | 0.0009 |
| *MMP3, MMP9, FBN2, ITGB, AURKB, SPDEF* | | | | |
| **135a-5p** | 0.94 | 0.024 | 0.84 | 0.014 |
| *RUNX2* | | | | |
| **9-5p** | 1.38 | 0.004 | 1.11 | 0.12 |
| *JAK3, FOXG1, ONECUT2* | | | | |

**miRNA upregulated in ccrcc1_4 tumors**

| | log2(FC) | FDR | HR(OS) | FDR |
|---|---|---|---|---|
| **370-3p** | -1.86 | 2.9e-6 | 1.17 | 0.025 |
| **134-5p** | -1.61 | 7.9e-7 | 1.27 | 0.025 |
| **409-3p** | -1.60 | 2.9e-6 | 1.22 | 0.0495 |
| **1307-3p** | -1.37 | 0.00001 | 1.31 | 0.0295 |
| **182-5p** | -1.22 | 8.6e-6 | 1.25 | 0.025 |
| **335-3p** | -1.20 | 0.0001 | 1.26 | 0.025 |
| **224-5p** | -1.19 | 0.001 | 1.24 | 0.0498 |
| *KLK1* | | | | |
| **193b-3p** | -1.19 | 0.0003 | 1.28 | 0.007 |
| **574-5p** | -1.14 | 8.6e-6 | 1.44 | 0.016 |
| **193b-5p** | -1.08 | 0.009 | 1.27 | 0.012 |
| **146a-5p** | -1.05 | 0.0004 | 1.17 | 0.2 |
| *ATOH8, C8A, IL12RB2* | | | | |
| **3529-3p** | -1.03 | 0.00009 | 1.39 | 0.013 |
| **34c-5p** | -1.02 | 0.004 | 1.32 | 0.0004 |
| **96-5p** | -1.01 | 0.0005 | 1.17 | 0.12 |
| *AQP5, MYRIP* | | | | |
| **199b-5p** | -0.93 | 0.001 | 1.29 | 0.014 |
| **369-3p** | -0.90 | 0.002 | 1.24 | 0.0498 |
| **146b-5p** | -0.82 | 0.002 | 1.46 | 0.0004 |
| **652-3p** | -0.78 | 0.0007 | 1.37 | 0.039 |
| **149-5p** | -0.78 | 0.001 | 1.38 | 0.025 |
| **1301-3p** | -0.77 | 0.0007 | 1.42 | 0.032 |
| **21-3p** | -0.75 | 0.005 | 1.49 | 0.0004 |
| **629-5p** | -0.73 | 0.006 | 1.36 | 0.025 |

*(Continued)*

**Table 2.** (Continued)

| 185-5p | -0.68 | 0.003 | 1.41 | 0.016 |
|---|---|---|---|---|
| let-7i-5p | -0.64 | 0.0004 | 1.70 | 0.002 |
| | *TRIM71* | | | |
| 320a | -0.63 | 0.017 | 1.28 | 0.019 |
| 212-5p | -0.62 | 0.01 | 1.36 | 0.022 |
| 155-5p | -0.62 | 0.039 | 1.18 | 0.16 |
| | *AGTR1* | | | |
| 222-3p | -0.61 | 0.006 | 1.34 | 0.039 |
| 425-5p | -0.52 | 0.002 | 1.61 | 0.014 |
| 21-5p | -0.44 | 0.028 | 1.69 | 0.003 |

FC = fold change; FDR = false discovery rate; HR = hazard ratio: OS = overall survival since diagnosis.

throughout the entire dataset, miRNAs with relative upregulation in ccrcc1_4 were associated with worse OS, and miRNAs with relative upregulation in ccrcc2_3 correlated with better OS. Pearson correlation of subtype-specific expression and OS, which were quantified as log2(fold

**Table 3. Brief literature review of selected miRNAs with the highest discriminatory power for molecular subtype and OS.**

| **miRNAs that were relatively upregulated in ccrcc1 and -4 tumors and associated with poor OS** | |
|---|---|
| let-7i-5p | The let-7 family is expressed higher in RCC compared to normal kidney. In RCC cell lines, a let-7 inhibitor inhibits proliferation and induces apoptosis [27]. Let-7a is correlated with intermediate prognosis in ccRCC [4]. In other tumor types, let-7 family members are described as tumor suppressors targeting oncogenes like MYC and RAS-family [27,36]. |
| 21 | Well-known oncomiR [3]. It is associated with poor outcomes in RCC. Inhibition results in cell cycle arrest, apoptosis and reduced invasive capacity [4,22–24,37]. |
| 425-5p | OncomiR [38,39]. Associated with poor survival in ccRCC treated with sunitinib and in chromophobe RCC [23,39]. Upregulation in RCC cell lines promotes invasiveness and inhibited apoptosis [28]. |
| 146b-5p | Highly expressed by regulatory T-cells (which are mostly found in ccrcc4 tumors). Antagomirs suppress $T_{regs}$ in vitro and in vivo [40]. The related miR-146a is hypoxia-regulated and promotes ccRCC proliferation and invasion [41]. |
| 34c-5p | The miR-34 tumor suppressor family are direct targets of p53, they repress cell proliferation and induced apoptosis. In RCC however, higher expression of miR-34a is reported in non-responders to sunitinib and in early progressors. In colon carcinoma higher expression of miR-34-b/c is reported in aggressive disease [7,26,42,43]. |
| 1307-3p | Serum levels are higher in patients with breast cancer, compared to those without [44]. |
| 134-5p | Tumor suppressing role in various cancer types [45,46]. |
| 193b-5p | Tumor suppressing role in various cancer types. In RCC, higher expression is associated with higher stage and early relapsing disease [26]. |
| **miRNAs that were relatively upregulated in ccrcc2 and -3 tumors and associated with long OS** | |
| 204-5p | miR-204 suppresses mRNA targets that otherwise promote tumor invasiveness and metastases. 204-5p expression is lower in the TCGA poor-prognostic miRNA cluster. miR-204 suppresses ccRCC tumor growth by controlling oncogenic autophagy. Expression is higher in non-metastatic compared to metastatic ccRCC [4,5,18]. |
| 135a-5p | miR-135a is a tumor suppressor in RCC, that induces tumor cell apoptosis and represses metastasis [47–49]. |
| 30c-5p | The miR-30 family acts as tumor suppressor in various cancer types. miR-30c is upregulated in non-metastatic compared to metastatic ccRCC and inhibits epithelial-to-mesenchymal transition [5,20]. |

OncomiR = miRNA that is associated with cancer. AntagomiR = synthetic RNA that is perfectly complementary to a specific miRNA target.

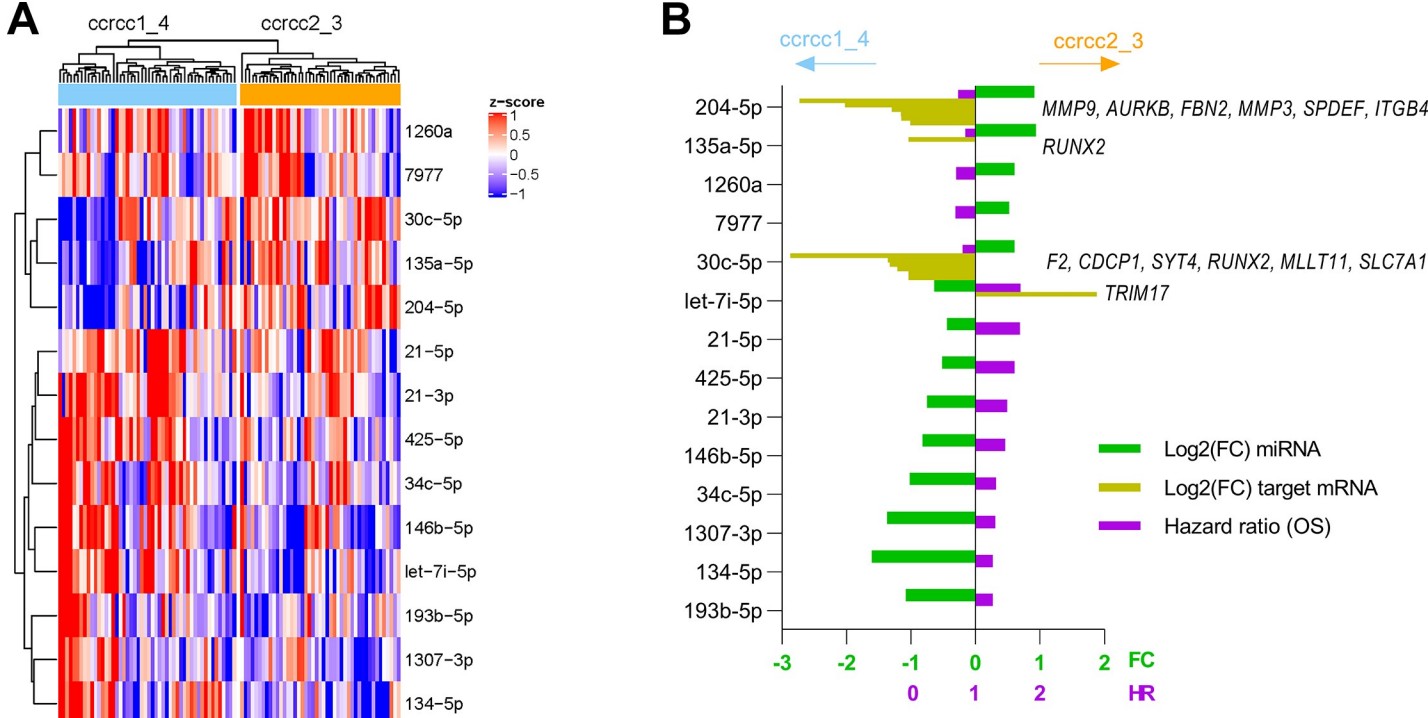

**Fig 3. Summary of selected miRNAs with the highest discriminatory power for both molecular subtype and OS (Leuven dataset, n = 95).** All miRNAs that were associated with poor prognostic ccrcc1_4 subtypes were systematically associated with poor OS, and *vice versa*. **(A)** Heatmap. **(B)** Bar plots quantifying subtype-specific miRNA expression, hazard ratio for OS and subtype-specific mRNA expression of predicted mRNA targets that were downregulated with a fold change ≤ 0.5 compared to the other subtype. All results are significant after correction for multiple testing (FDR < 0.05). FC = fold change; OS = overall survival.

change) and hazard ratio respectively, showed a very robust correlation with r = -0.54 (95% CI (-0.61,-0.46) and p < 0.0001) (Fig 4). This indicates a general shift of miRNA expression between less and more aggressive tumors, rather than only a limited subset of miRNAs being associated with aggressive disease. An overview of miRNAs that were significantly correlated with both molecular subtype and OS is provided in Table 2. Results were similar when we repeated outcome analysis using time to metastasis and OS since metastasis, as well as after randomly dividing the cohort in two groups, which underscores the validity of these results (S3 Fig and S1 Table).

To test whether individual miRNAs have independent prognostic value, we performed multivariate Cox regressions with the well-established IMDC risk groups and with molecular subtype. Of the 14 miRNAs with the highest discriminatory power for both molecular subtype and OS (Fig 3 and Table 3), all but miR 30c-5p remained independent predictors for OS after diagnosis (S2 Table). Molecular subtype remained an independent predictor as well, except when tested against miRs 21-3p (upregulated in unfavorable ccrcc1_4) and 204-5p (upregulated in favorable ccrcc2_3). This is in line with the TCGA miRNA clusters, where the poor prognosis miRNA cluster could be discriminated based on upregulation of 21-3p together with downregulation of 204-5p [4].

## MiRNA expression is not predictive for response to sunitinib or pazopanib

The majority of patients (77%) were treated in first line with the anti-angiogenic agents sunitinib or pazopanib. These agents were standard first-line therapies for advanced ccRCC before the introduction of immune checkpoint inhibitors, and are currently routinely used from

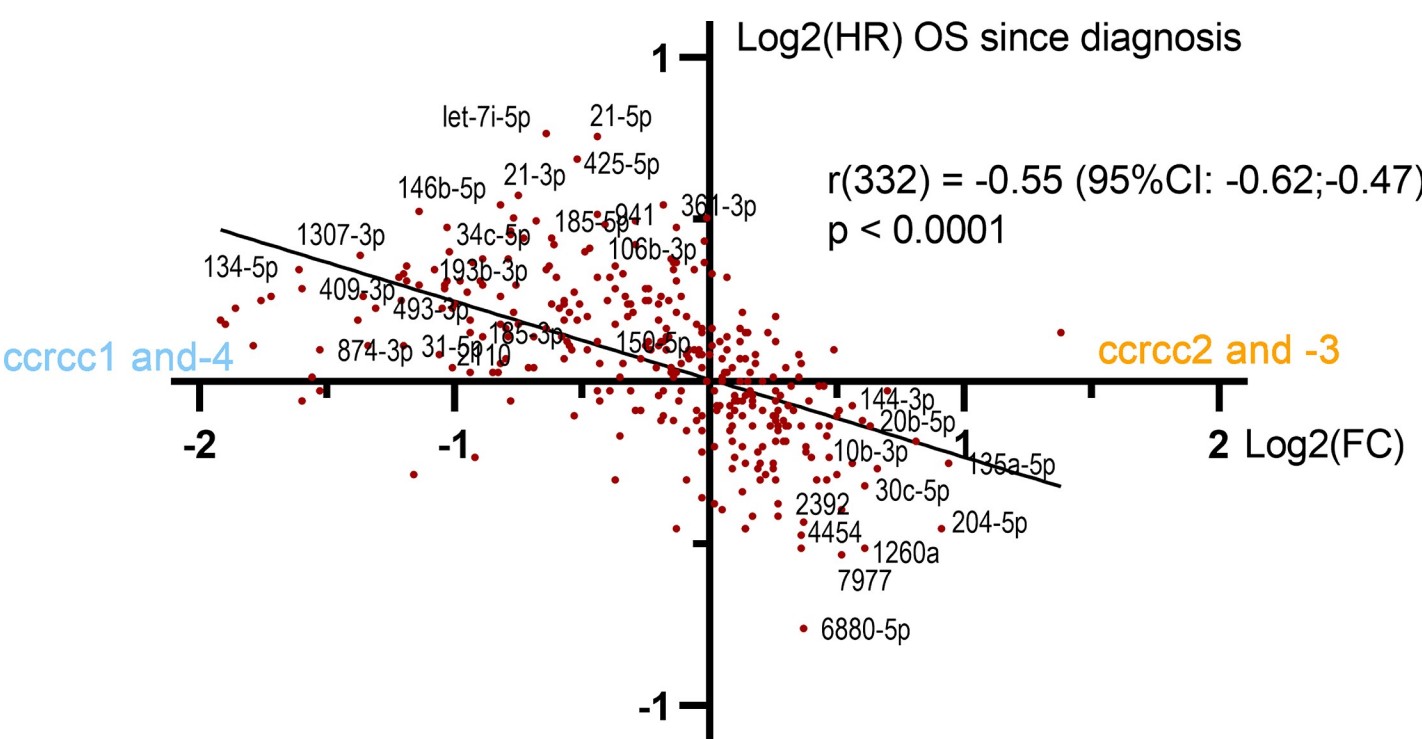

**Fig 4. Scatterplot illustrating the correlation between subtype-specific miRNA expression (Log2(FC)) and HR for OS (Leuven dataset, n = 95).** There was a strong correlation throughout the dataset, with a Pearson coefficient of -0.54 (332 degrees of freedom). Hazard ratio for OS is plotted on a log2-scale for graphical reasons; the regression line was fitted on the original data. HR = hazard ratio; OS = overall survival; FC = fold change.

second line on [16]. To assess whether specific miRNAs could be predictive for response to sunitinib or pazopanib, we tested their association with response rate and progression free survival. No miRNAs were significantly associated with response rate after correction for multiple testing, and only three were significantly associated with progression free survival (miRs 185-5p, 193b-5p and 193b-3p). As these latter were also strongly associated with OS since diagnosis, we deemed them to be prognostic rather than predictive.

## Discussion

In this study, we analyzed miRNomes of ccRCC to assess the link between miRNA expression and transcriptomic molecular subtypes, mRNA expression and clinical outcomes.

The ccrcc1 to -4 molecular subtypes showed clearly different miRNA expression patterns, both in our dataset and the TCGA validation set. First, unsupervised clustering of miRNAs revealed two clusters that overlapped with the favorable (ccrcc2_3) and unfavorable (ccrcc1_4) subtypes. This is in line with earlier findings from TCGA, where four miRNA clusters were found, of which only two overlapped with molecular subtypes [4]. Secondly, a supervised analysis revealed several miRNAs that were discriminatory between the subtypes. Moreover, miRNAs that were expressed higher in unfavorable subtypes were systematically associated with poor OS, and miRNAs that were expressed higher in favorable subtypes with short OS. This held true in the entire dataset, and not only for those miRNAs with significant subtype-specific expression (Fig 4, S1 Table). These findings underscore the robustness of these subtypes.

A striking finding, which was not expected at study set-up, was the strong correlation between subtype-specific miRNA expression and OS, throughout the entire dataset. Put otherwise, there was a shift in global miRNA expression between less *vs* more aggressive tumors.

This is in contrast to published literature, which typically describes small subsets of miRNAs that are significantly associated with prognosis in some studies but not in others. Our study reveals that aggressive tumors are not set apart by only a few distinctive miRNAs. Rather, these tumors display an entirely different miRNome expression, regardless of single miRNAs reaching the significance threshold for OS. Importantly, this is the case in both advanced and earlier stage disease, as findings were similar in our dataset (stage IV ccRCC) and the TCGA (mostly stage I-III ccRCC).

Many of the most discriminatory miRNAs for both molecular subtype and OS are well described oncomiRs or tumor suppressors in ccRCC or other tumor types. Previously, two miRNA signatures have been published that discriminated localized from metastatic ccRCC [5,9]. According to these signatures, miRs 10b, 139-5p, 30a-3p, 30c-5p, 139-5p and 144-5p were discriminative for localized disease, and miRs 130b and 199b-5p for metastatic tumors. Our series validated the prognostic value of all these miRs, with the former all being associated with ccrcc2_3 and long OS, and the latter with ccrcc1_4 and short OS. As we only used biological material from metastatic tumors we could not validate the capability of these signatures to predict the risk of relapse.

When correlating miRNA expression levels with their predicted mRNA targets, several subtype-specific miRNAs repressed targets that are involved in tumor epithelial-to-mesenchymal transition. Tumors with a mesenchymal phenotype have an aggressive behavior and poor prognosis. In RCC, they are best represented by the notoriously unfavorable sarcomatoid RCC, which are typically of the unfavorable ccrcc4 subtype [10,17]. In the favorable ccrcc2_3 subtypes, miR-204-5p was strongly upregulated, predicted long OS, and repressed several targets that are involved in tumor invasiveness and metastasis (*MMP3*, *MMP9*, *AURKB*, *FBN2*, *ITGB4*, *SPDEF*). miR-204 is a well-studied tumor suppressor that inhibits epithelial-to-mesenchymal transition, as well as the immune suppressive interleukin 11 [8,18]. In ccRCC, miR-204 has been shown to suppress tumor growth by controlling oncogenic autophagy, and it is expressed at higher levels in non-metastatic ccRCC compared to metastatic disease [5,19]. Indeed, in the TCGA cohort, the poorest prognosis miRNA cluster showed downregulation of miR-204 [4]. Expression of miR-30 family members, and especially 30c-5p, was also markedly favorable. The members of this family have a well-established role as tumor suppressors inhibiting epithelial-to-mesenchymal transition in various cancer types [20]. In our series, it repressed targets involved in invasiveness and metastasis (*F2*, *RUNX2*, *CDCP1*). Earlier studies in ccRCC showed that miR-30c inhibits epithelial-to-mesenchymal transition and reduces HIF2A activity, which blocked xenograft tumor growth [21,22]. It is expressed at higher levels in non-metastatic compared to metastatic ccRCC [5]. Another miRNA that was expressed higher in ccrcc2_3 tumors and showed a trend towards good prognosis is miR-10b. miR-10b was discriminatory for the good-prognosis miRNA cluster in the TCGA cohort, is part of two prognostic ccRCC signatures and is expressed higher in non-metastatic ccRCC [4,5,9]. On the other hand, when looking at unfavorable miRNAs, miR-21 was strongly correlated with ccrcc1_4 subtypes and poor OS. MiR-21 is indeed one of the best described oncomiRs and may contribute to the general downregulation of miRNAs in cancer [3]. This was also described by TCGA, where miR-21 was discriminative for the miRNA cluster with poorest prognosis and repressed targets in the VHL–HIF1A axis [4]. Zaman et al. showed that miR-21 was increased in high stage tumors and that its inhibition in ccRCC cell lines led to cell cycle arrest, apoptosis and reduced invasive and migratory capacities; other groups have reported similar findings [23–26].

Counterintuitively, some of the miRNAs that are most discriminatory for the unfavorable ccrcc1_4 subtypes and poor OS, are well known tumor suppressors: let-7i-5p, 34c-5p, 134-5p and 193-5p. miRNA regulation is indeed so complex, that contradictory results are often

reported for even the best established miRNA pathways [2]. One specific hypothesis on the current results is that downregulation or loss of these tumor suppressive miRs may act as a key oncogenic mechanism in the ccrcc2_3 subtypes, whereas ccrcc1_4 subtypes depend on other oncogenic pathways. This is supported by the finding that the expression of miR-34c-5p, 134-5p and 193-5p is lower in ccrcc2_3 tumors compared to normal kidney, suggesting active loss (Fig 1D). Moreover, miR-34c and 193b were also associated with early progressive disease in ccRCC in other series [7]. Let-7i, although part of the well-established let-7 tumor suppressor family, was strongly associated with ccrcc1_4 tumors and poor OS. This is in line with a previous study reporting that a let-7 inhibitor caused apoptosis in ccRCC cell lines, and with the TCGA where let-7a was correlated with intermediate prognosis [4,27].

We did not identify any miRNAs that were predictive for response to the anti-angiogenic agents sunitinib or pazopanib in 99 patients that received first-line treatment with these therapies. These findings are consistent with other studies, that only described a correlation of miR-NAs with progression free survival or OS, but not response rates [6,7,28].

While our miRNome-wide approach was indispensable to reveal the general shifts in miRNA expression between less and more aggressive tumors, it also constitutes a limitation of this study. We did not assess single miRNAs that may be fundamentally involved in the regulation of angiogenic or immune pathways, which constitute therapeutic targets in ccRCC, and can therefore not exclude that some predictive miRNAs may exist which did not survive correction for multiple testing. We did not seek to validate the effects of discriminative miRNAs with functional studies either, but such studies have been done previously for most of the most discriminative miRNAs.

## Conclusion

The ccrcc1 to -4 transcriptomic molecular subtypes in ccRCC have fundamentally different miRNA expression profiles that underlie their mRNA expression, which underscores the robustness of these subtypes on a multi-omics scale. There was a very strong association between subtype-specific miRNA expression (favorable *vs* unfavorable) and outcome (long *vs* short OS), throughout the entire dataset. This reflects a general shift in miRNA expression from less to more aggressive tumors. Many of the discriminative miRNAs repressed mRNA targets that are involved in tumor epithelial-to-mesenchymal transition.

## Methods

### Patients and samples

We selected primary ccRCC samples from the Leuven University Hospitals ccRCC tumor bank, which prospectively collects tissue and detailed clinical data from metastatic ccRCC patients after informed consent. Patients were eligible if archived tissue of the primary tumor was available, provided they had not received systemic therapy prior to tissue retrieval. Clear-cell histology was confirmed by an expert uro-pathologist. As a reference, we included paired normal kidney samples from 16 patients as well (healthy tissue block from the nephrectomy specimen removed during ccRCC surgery). The protocol for miRNome analysis has been described previously [6]. Briefly, total RNA was extracted from formalin-fixed paraffin-embedded (FFPE) tissue using the Recover All Total Nucleic Acid Isolation kit for FFPE (Ambion). Libraries were constructed with the NEBNext Multiplex Small RNA Library Prep Set for Illumina (New England Biolabs E7300) and sequenced for 50 bases in a single-read format (Genome Analyzer Ilx, Illumina). In cases where fresh frozen primary tumor tissue was available, this was used to determine ccrcc1 to -4 transcriptomic molecular subtype, using the established 35-gene classifier algorithm as described previously [10]. The TCGA ccRCC cohort

was used as a validation set. In TCGA samples for which combined miRNome and transcriptome data derived from fresh frozen tissue were available, we determined molecular subtype using the classification algorithm.

## Small RNAseq preprocessing

Raw reads were trimmed using Trim Galore! (Babraham Bioinformatics) and aligned against the miRbase mature miRNA sequences using Bowtie1 (Johns Hopkins University), resulting in raw read counts for 2589 miRNAs [29]. For the clustering and differential expression results, raw count data was analyzed according to the DESeq2 workflow (Bioconductor) after filtering out miRNA with a normalized count smaller than 5 in more than 50 samples [30]. For the correlation with overall survival (OS) since diagnosis of ccRCC, counts were normalized using the upperquartile method as implemented in the R package edgeR after removing lowly expressed miRNAs (zero counts per million reads in more than 100 samples) [31].

## Unsupervised clustering

To assess the presence of miRNA clusters and their overlap with previously determined mRNA molecular subtypes, we performed unsupervised clustering. For this, a principal component analysis was run on variance stabilized counts obtained from DESeq2 (version 1.25.16). In addition, the samples were clustered hierarchically using the 50 most variable miRNAs using correlation as a distance metric as implemented in the R package ComplexHeatmap (version 2.1.1, Bioconductor) [32]. To further investigate the miRNA based clustering, variance stabilized counts of all expressed miRNAs were normalized and tumor samples were partitioned into two clusters according to the 'partitioning around medoids' algorithm as implemented in the R package cluster (version 2.1.0). A Chi-Square test was performed to assess the association of the favorable and unfavorable prognosis mRNA subtypes with the two miRNA clusters.

## Differential expression

Differential miRNA expression between the favorable (ccrcc2_3) *versus* (*vs*) the unfavorable prognostic subtypes (ccrcc1_4) and tumor *vs* normal samples was assessed using DESeq2 (version 1.25.16). As a validation of the differential expression results, miRNA data from the TCGA ccRCC cohort was analyzed in a similar fashion using DESeq2.

## Functional characterization

For miRNAs with subtype-specific expression (defined by log2 (fold change) $\geq$ 0.5 or $\leq$ -0.5 and false discovery rate (FDR) < 0.05), Ingenuity Pathway Analysis (IPA, Qiagen) was used to collect the predicted downstream mRNA targets. Moreover, this data was integrated with the subtype-specific expression of mRNA targets in the TCGA dataset to detect putative miRNA-mRNA functional interactions (microRNA Target Filter module, IPA).

Kyoto Encyclopedia of Genes and Genomes (KEGG) gene set enrichment analysis on the differentially expressed miRNAs was performed using the R package RBiomirGS (version 0.2.12) [33]. Briefly, the miRNA differential expression list is used to search multiple databases for miRNA-mRNA interactions, resulting in a target mRNA list. For each miRNA, the fold change and p-value from the miRNA differential expression list are used to calculate an expression score. From this, a miRNA impact score for the target mRNAs is generated. With the mRNA score and a gene set database, gene set enrichment is calculated using logistic regression.

## Correlation with clinical outcome

We performed univariate Cox regression to correlate individual miRNAs with OS since diagnosis and progression free survival on first-line sunitinib or pazopanib. We used a Mann-Whitney U test to analyze expression differences between patients with partial response *versus* progressive disease at first evaluation. P-values were corrected for multiple testing using the Benjamini and Hochberg method, and are reported as FDR [34]. In another effort to confirm the validity of these results, we also performed Cox regression for other outcome parameters (OS since metastasis and time to metastasis). Lastly, as internal validation, we repeated all analyses after randomly dividing our patient population into two cohorts.

In a selection of miRNAs with the strongest association with both molecular subtype and OS, we performed multivariate Cox regressions with International Metastatic ccRCC Database Consortium (IMDC) risk groups and molecular subtype [35].

Molecular analyses were done with R version 3.5.1, clinical outcome analyses with SAS version 9.4. All statistical tests were two-sided.

## Study approval

The study was approved by the Ethics Committee Research UZ / KU Leuven. Patients whose tissue was used had given written informed consent. In some cases, we used biological material from deceased patients for whom a general positive advice was given by the Ethical Committee for the use of remaining tissue for research purposes.

## Supporting information

**S1 Table. Log2(fold change) for ccrcc2_3 *vs* ccrcc1_4 for every miRNA, as well as hazard ratios for OS after diagnosis, OS since metastasis and time to metastasis, progression free survival on first-line sunitinib or pazopanib and Mann-Whitney test for response *versus* progressive disease.**
(XLSX)

**S2 Table. Multivariate Cox regressions for overall survival after diagnosis, testing selected miRNAs against International Metastatic ccRCC Database Consortium (IMDC) risk groups and molecular subtype.** All miRs except for miR 30c-5p remained independent predictors for OS. Molecular subtype remained and independent predictor as well, except when tested against miRs 21-3p (up in unfavorable ccrcc1_4) and 204-5p (up in favorable ccrcc2_3). IMDC risk groups: good (G), intermediate (I) and poor (P).
(DOCX)

**S1 Fig. Clustering on TCGA samples.** Clustering reveals two miRNA clusters with 66% and 91% overlap with favorable ccrcc2_3 and unfavorable ccrcc1_4 molecular subtypes respectively (p = 4.7e-9).
(PDF)

**S2 Fig. KEGG gene set enrichment analysis on subtype-specific miRNAs.** KEGG analysis showed less suppression of pathways involved in retaining an epithelial phenotype in ccrcc2_3 tumors, suggesting less possibility for epithelial-to-mesenchymal transition in these tumors. The biological interpretation of the model coefficient can be stated as follows (in the context of two-group comparison, i.e., ccrcc2_3 vs ccrcc1_4): if the coefficient is positive, miRNA inhibition on target mRNAs might be lifted, thereby leading to less suppression on the gene set of interest in the experimental group.
(PDF)

**S3 Fig.** Differences in miRNA expression were similar across outcome parameters: OS since diagnosis (main text), OS since stage IV (A) and time to metastases (B). HR = hazard ratio; FC = fold change.
(TIF)

## Acknowledgments

The authors would like to thank Maria Santos for her excellent support in small RNA library preparation and the French National League Against Cancer for their initial co-development of the ccrcc1 to -4 molecular subtypes.

## Author Contributions

**Conceptualization:** Annelies Verbiest, Stefano Caruso, Benoit Beuselinck.

**Data curation:** Annelies Verbiest, Vincent Van Hoef, Jesús García-Donas, Osvaldo Graña-Castro, Marcella Baldewijns.

**Formal analysis:** Annelies Verbiest, Vincent Van Hoef, Osvaldo Graña-Castro, Annouschka Laenen.

**Methodology:** Annelies Verbiest, Vincent Van Hoef.

**Project administration:** Benoit Beuselinck.

**Resources:** Jesús García-Donas, Maarten Albersen, Eduard Roussel, Benoit Beuselinck.

**Software:** Vincent Van Hoef, Osvaldo Graña-Castro, Stefano Caruso.

**Supervision:** Cristina Rodriguez-Antona, Patrick Schöffski, Agnieszka Wozniak, Jessica Zucman-Rossi, Benoit Beuselinck.

**Visualization:** Annelies Verbiest, Vincent Van Hoef.

**Writing – original draft:** Annelies Verbiest, Vincent Van Hoef.

**Writing – review & editing:** Annelies Verbiest, Cristina Rodriguez-Antona, Patrick Schöffski, Agnieszka Wozniak, Stefano Caruso, Gabrielle Couchy, Jessica Zucman-Rossi, Benoit Beuselinck.

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
