## [Decision Letter · Decision Letter 0]

4 Jun 2020

PONE-D-20-04595

MicroRNA expression profiles in molecular subtypes of clear-cell renal cell carcinoma are associated with clinical outcome and repression of specific mRNA targets.

PLOS ONE

Dear Dr. Annelies Verbiest,

Thank you for submitting your manuscript to PLOS ONE. After careful consideration, we feel that it has merit but does not fully meet PLOS ONE’s publication criteria as it currently stands. Therefore, we invite you to submit a revised version of the manuscript that addresses the points raised during the review process.

Please ensure that all comments of the reviewer are fully addressed in the revised manuscript.

We look forward to receiving your revised manuscript.

Kind regards,

Olorunseun Ogunwobi, MD, PhD

Academic Editor

PLOS ONE

Journal Requirements:

2.  Please provide additional details regarding participant consent. In the ethics statement in the Methods and online submission information, please ensure that you have specified what type of consent you obtained (for instance, written or verbal, and if verbal, how it was documented and witnessed).

3. At this time, please provide the accession number of the data deposited in ArrayExpress (line 98).

'I have read the journal's policy and the authors of this manuscript have the following

competing interests: Benoit Beuselinck received consultancy fees from Amgen, Ipsen,

Pfizer and Novartis and institutional research grants from Bristol-Myers Squibb and

Ipsen. Patrick Schöffski has received consultancy fees as well as institutional research

grants from Merck and Exelixis. The other authors have no conflicts of interest to

declare.'

'The authors would like to thank Maria Santos for her excellent support in small RNA library preparation,

the French National League Against Cancer for their initial co-development of the ccrcc1 to -4 molecular

subtypes, and the Kom Op Tegen Kanker (Stand up against cancer) foundation and Research Foundation

Flanders for their financial support'

'The author(s) received no specific funding for this work.'

Reviewers' comments:

Reviewer's Responses to Questions

**Comments to the Author**

1. Is the manuscript technically sound, and do the data support the conclusions?

Reviewer #1: Yes

2. Has the statistical analysis been performed appropriately and rigorously? 

Reviewer #1: Yes

3. Have the authors made all data underlying the findings in their manuscript fully available?

Reviewer #1: Yes

4. Is the manuscript presented in an intelligible fashion and written in standard English?

Reviewer #1: Yes

5. Review Comments to the Author

Reviewer #1: The question, study execution, and results are straightforward. Also, few technical issues are either not discussed or not addressed. The authors must carefully address all of my concerns before further consideration.

1. The figure quality should be improved.

2. Line number: 59

Please cite reference in proper place.

3. Line number: 116

Please provide the corresponding figure validated by tcga dataset.

4. Line number: 186

What is the full form of OS?

5. Line number: 215

Please provide a figure/ table to support “MiRNA expression is not predictive for response to sunitinib or pazopanib”

6. PLOS authors have the option to publish the peer review history of their article (what does this mean?). If published, this will include your full peer review and any attached files.

Reviewer #1: No

---

## [Author Response · Author response to Decision Letter 0]

22 Jul 2020

The manuscript has been adapted to all editor and reviewer comments, as can be found in the track changes and "response to reviewers" document. 

Reviewer #1: The question, study execution, and results are straightforward. Also, few technical issues are either not discussed or not addressed. The authors must carefully address all of my concerns before further consideration.

We thank the reviewer for his/her appreciation of the manuscript. 

1. The figure quality should be improved.

The figure quality is indeed greatly reduced in the pdf file for peer review. The separate figure files are high quality and adhere to Plos One requirements, including a quality check by PACE.

2. Line number: 59

Please cite reference in proper place.

The authors are not entirely sure what is meant by this comment? The reference is cited between brackets at the end of the phrase where it is first mentioned (we have opted to systematically cite at the end of sentences and not in the middle). 

3. Line number: 116

Please provide the corresponding figure validated by tcga dataset.

We have included it as Supporting Figure 1.

4. Line number: 186

What is the full form of OS?

We thank the reviewer for noticing that the abbreviation was incorrectly explained only in figure/table legends. We added “overall survival (OS)” at its first appearance in the text.

5. Line number: 215

Please provide a figure/ table to support “MiRNA expression is not predictive for response to sunitinib or pazopanib”

The results have been provided in Supporting Table 1.

---

## [Decision Letter · Decision Letter 1]

25 Aug 2020

MicroRNA expression profiles in molecular subtypes of clear-cell renal cell carcinoma are associated with clinical outcome and repression of specific mRNA targets.

PONE-D-20-04595R1

Dear Dr. Annelies Verbiest,

We’re pleased to inform you that your manuscript has been judged scientifically suitable for publication and will be formally accepted for publication once it meets all outstanding technical requirements.

Kind regards,

Olorunseun Ogunwobi, MD, PhD

Academic Editor

PLOS ONE

Reviewers' comments:

Reviewer's Responses to Questions

**Comments to the Author**

1. If the authors have adequately addressed your comments raised in a previous round of review and you feel that this manuscript is now acceptable for publication, you may indicate that here to bypass the “Comments to the Author” section, enter your conflict of interest statement in the “Confidential to Editor” section, and submit your "Accept" recommendation.

Reviewer #1: All comments have been addressed

2. Is the manuscript technically sound, and do the data support the conclusions?

Reviewer #1: Yes

3. Has the statistical analysis been performed appropriately and rigorously? 

Reviewer #1: Yes

4. Have the authors made all data underlying the findings in their manuscript fully available?

Reviewer #1: Yes

5. Is the manuscript presented in an intelligible fashion and written in standard English?

Reviewer #1: Yes

6. Review Comments to the Author

Reviewer #1: The authors have addressed all the technical questions and improved the quality of the figures. The manuscript can be considered for publication in PLOS ONE.

7. PLOS authors have the option to publish the peer review history of their article (what does this mean?). If published, this will include your full peer review and any attached files.

Reviewer #1: No

---

## [Editor Report · Acceptance letter]

31 Aug 2020

PONE-D-20-04595R1 

MicroRNA expression profiles in molecular subtypes of clear-cell renal cell carcinoma are associated with clinical outcome and repression of specific mRNA targets. 

Dear Dr. Verbiest:

I'm pleased to inform you that your manuscript has been deemed suitable for publication in PLOS ONE. Congratulations! Your manuscript is now with our production department. 

Kind regards, 

on behalf of

Dr Olorunseun Ogunwobi 

Academic Editor

PLOS ONE